# A Study on the Development of ICT Convergence Smart ESP Using Embedded System

**Joon-Ho Cho**

Department of Electronics Convergence Engineering, Wonkwang University, 460 Iksandae-ro, Iksan 54538, JeonBuk, Republic of Korea; cho1024@wku.ac.kr

**Abstract:** In this paper, the electrical submersible pump (ESP) is combined with information and communication technologies (ICT) to diagnose the operation status and soundness of the ESP. The ICT convergence provides users with maintenance and repair information through history management and remote control in case of failure. The proposed system includes a monitoring unit (MU) that senses the operating state of the ESP, a monitoring control unit (MCU) that transmits the sensed information to internal and external databases, and a monitoring system that allows users to check the status information. The server and embedded system can monitor the operation status of the submersible pump by storing sensor values in the database and displaying them on the screen. The embedded system retrieves the sensor values stored in the database and performs individual and complex diagnoses. The diagnosis results are sent to the server and status information to the monitoring control unit. The diagnosis of the submersible pump was divided into 23 individual sensor classifications, and a multilayer perceptron was implemented. Weights were set through learning and testing. The proposed ICT-converged Smart ESP is considered to be competitive as it greatly improves the existing system in terms of status and maintenance.

**Keywords:** electrical submersible pump; ICT; embedded system; monitoring system; diagnosis; neural network; motor

## 1. Introduction

Pumps, which account for approximately 20% of industrial power consumption, play a crucial role in various industries, the private sector, and the public sector. It is vital to ensure their normal operation and maintenance for energy efficiency and performance. If pump failures result in reduced efficiency or the cessation of operation, it can lead to significant industrial losses or even disasters, such as flood drainage pump failures. Additionally, with advances in technology, pumps have become larger, faster, and more electronic. Backup pumps are often no longer needed, which results in not only the need for repair time and monetary losses but also production losses in the event of an accident or breakdown.

Centrifugal pumps, widely used in industrial settings, experience damage in their hydraulic or mechanical parts. The main components of the pump, such as the bearings and impellers, are susceptible to various defects that greatly impact pump performance and lead to failure. This type of failure occurs over time and can be remedied by replacing parts. However, conventional methods cannot address unexpected failures. Thus, it is crucial to continuously monitor the pump operation status using the latest ICT technology to solve the problem.

The early detection of faults and an accurate diagnosis can enable fast maintenance, reduce costs, and extend the pump life [1,2]. The pump's operating state can be monitored by sensing vibrations, sound, pressure, temperature, humidity, voltage, and current, but this is often limited by increased cost and space restrictions. To overcome these constraints, embedded systems and ICT technologies have been developed to monitor submersible pump conditions. For instance, methods have been developed to detect cavitation phenomena in centrifugal pumps through motor current signature analysis [3] and to extract defects

in impellers using MSB (modulation signal bispectrum) [4]. The pump's current signal can provide information on complex non-linear processes, including various faults that typically exhibit non-stationary characteristics. Fourier spectrum analysis has a disadvantage of "spectral leakage" if the frequency resolution of the current spectrum is not sufficient, but much research has been performed on the time–frequency analysis method, including wavelet transform, which provides full and partial characteristics of a signal in both time and frequency domains. Additionally, many recent studies have been conducted on methods to analyze complex signals [5–9]. These methods characterize signals based on the analytic properties of the data without using basic functions, such as spectral and wavelet transforms. Research on failure diagnosis and prediction is continuously being conducted.

Various methods have been developed including: measuring the load on the motor to check its rotation state [10]; using the LSTM (long short-term memory) method to analyze motor acceleration, current, and vibration data [11,12]; combining LSTM with various theories, such as the wavelet [13], ant colony [14], recurrence quantification [15], and stacked convolutional bidirectional [16,17] methods; and using DRNN (deep recurrent neural network) [18]. Furthermore, real-time fault diagnosis technology has been extensively studied [19].

Previous studies diagnosed pump failures based on the value of a single sensor, while this paper added a composite diagnosis method and applied a multi-perceptron to predict failures.

In this paper, we propose an ICT convergence smart ESP that integrates ICT technology with ESP to acquire, analyze, and diagnose sensor information. The monitoring unit (MU) and monitoring control unit (MCU) hardware are designed to check ESP operation status and transmit information, while the software (monitoring and diagnosis) is developed to check and diagnose status information. Our proposal suggests that the failure diagnosis of ESP can be conducted through complex diagnosis instead of relying solely on the individual sensor values. This paper is structured in the following order: hardware specifications and functions for fault diagnosis, development of the monitoring system, fault diagnosis algorithm through complex diagnosis, and the conclusion.

## 2. ICT Convergence ESP System Development

In this paper, we propose an embedded system-based ICT convergence smart ESP monitoring system that integrates power and signal cables. Our system applies technology to combine power and signal cables, which were previously separated, and transmits pre-processed signals from the ESP to the embedded system via RS485 communication.

In this paper, an embedded system-based ICT convergence smart ESP monitoring system is presented. The system applies technology to integrate power and signal cables, transmit pre-processed signals from the ESP to the embedded system through RS485 communication, and apply a complex diagnosis algorithm to analyze pump information for normal operation and dangerous conditions. Additionally, the embedded integrated board designed inside the ESP applies internal electromagnetic shielding technology, and the cable that transmits sensor signals also uses electromagnetic shielding cables to increase communication stability. Information on the ESP's status is displayed on the screen and a warning sound is generated in case of any dangerous conditions. Moreover, system control technology is applied to contact the pump operator in case of an emergency. Figure 1 shows the block diagram of the proposed ICT convergence smart ESP monitoring system.

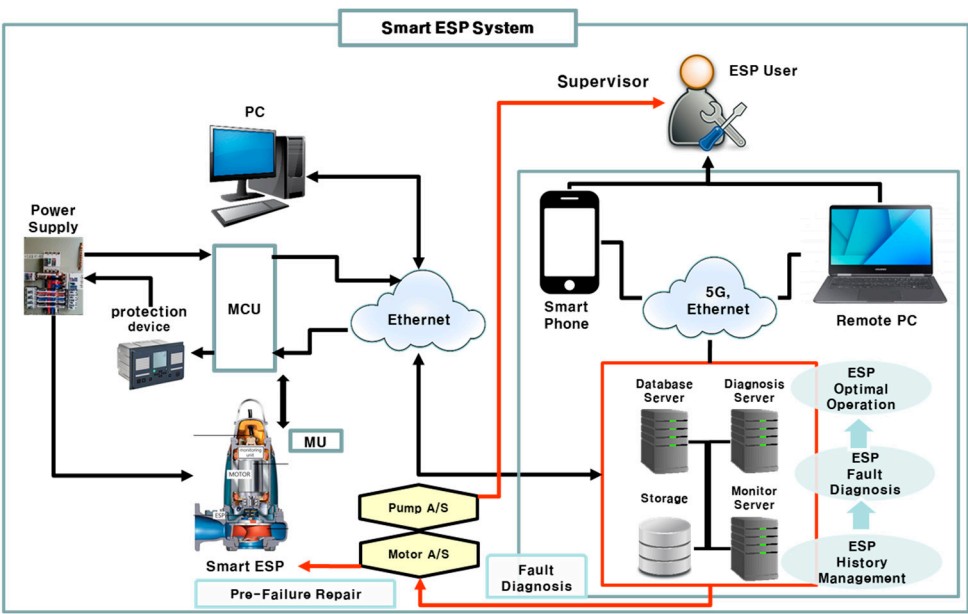

**Figure 1.** Schemes follow the same formatting.

## 2.1. Submersible Pump Monitoring Unit Design

The monitoring unit of the submersible pump acquires signals from sensors attached to monitor the condition of the ESP's motor and pump. Additionally, the operating state of the ESP can be checked based on the information obtained from each terminal sensor, and it can communicate with the MCU attached to the top. This paper uses various types of sensors to check the ESP status, including temperature, water leakage, slope, pressure, vibration, humidity, voltage, current, and rotational speed. Figure 2 shows the block diagram of the MU.

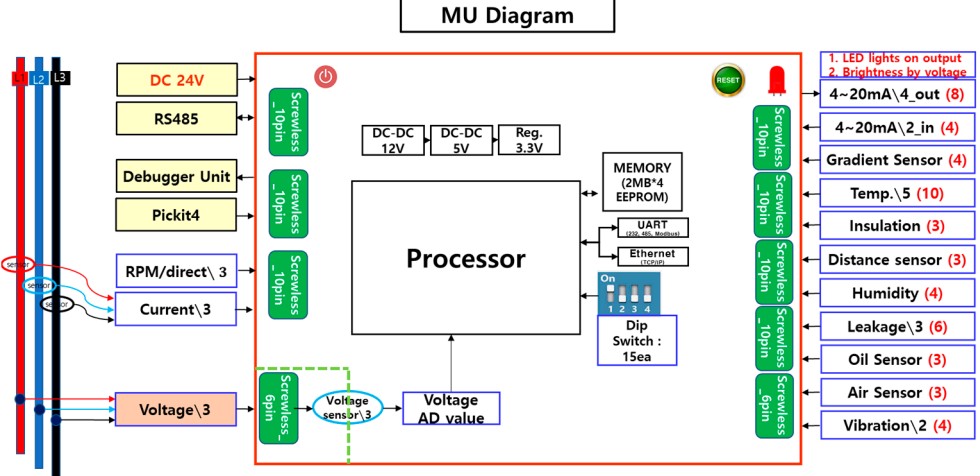

**Figure 2.** MU block diagram.

The temperature sensors listed in Table 1 were installed on the upper and lower parts of the windings (L1, L2, L3) and bearings, and the water leakage sensor was installed on the upper and lower parts as well. The inclination sensor must be initialized based on the installed position of the submersible pump. Vibration sensors were installed at the top and bottom, and humidity was installed at the center of the pump. Voltage and current were installed in each winding (L1, L2, L3), and the rotational speed and direction could be measured. Table 1 presents the types of sensors used and their measurement ranges.

**Table 1.** Tables should be placed in the main text near to the first time they are cited.

| | Division | Sensor Type | Measuring Range |
|---|---|---|---|
| Temperature | Winding (L1,L2,L3) | PT 100 (Line type) | −20~200 °C |
| | Bearing (up, down) | PT l00 (Volt type) | −20~200 °C |
| Leak | up, down | Self-production | Dry/Flooding |
| Gradient | | SINVT | x,y (−90~90 °C)/±0.1% |
| Pressure | Oil | CYYZ 31-A1-19 | 0~100 kpa (±0.25%FS) |
| | Air | PSS-01A-R1 | 0~100 kpa (±1%FS) |
| Oscillation | Bearing (up, down) | EVT-101-VR20-05 | 0~20 mm/s (±5%FS) |
| Humidity | | WTS3535 | 0~100 RH% (±3%FS) |
| Voltage (L1, L2, L3) | | Trans | 0~600 V |
| Electric Current | | CT | 0~120 A |
| Rotation Speed | | PR12-2DP | 0~7200 rpm |

(Microprocessor used ATSAMC21G17A of ARM series).

### 2.2. Design of Monitoring Control Unit of Submersible Pump

The MCU serves as an intermediate stage between the submersible pump, server, and embedded system. It transmits the sensor values of the submersible pump and directly performs operations according to the pump's diagnosis result.

Based on the information obtained from the MU, the MCU can drive the relay to operate the protection device or transmit the values to the monitor of the embedded system to provide detailed information to the manager. Additionally, it is transmitted to the embedded system directly connected to the server computer via TCP/IP communication. If the individual and complex diagnosis result indicates a faulty state, the relay is operated to stop the pump. Figure 3 is a block diagram of the MCU that shows the process of transferring sensor values. When the sensor value is transmitted to the server computer and embedded system, it is converted into an actual value and transmitted.

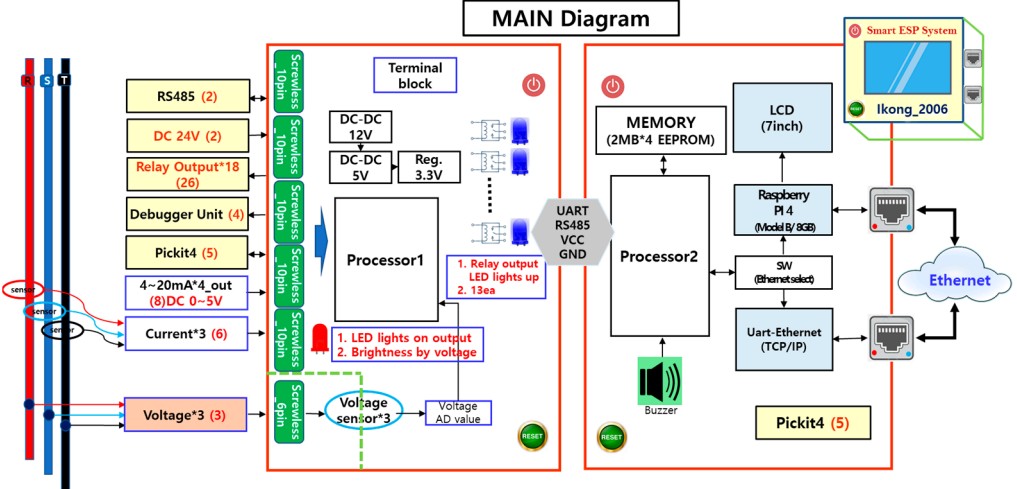

**Figure 3.** MCU block diagram.

### 2.3. Monitoring of Embedded System

The monitoring system of the embedded system is designed to check the sensor information from the MU on-site. It is necessary to have an on-site monitoring system in case of server data failure or the loss of the internet connection. The screen displays temperature, humidity, pressure, voltage, current, vibration, tilt, rotation, and both indi-

vidual and complex diagnosis results. The program was developed using Python and performs individual and complex diagnostics by reading the data values of the database. The screen is numerically displayed in three steps, similar to the server computer, with a red background indicating blocking, a blue background indicating caution, and a white background indicating normal status, as shown in Figure 4.

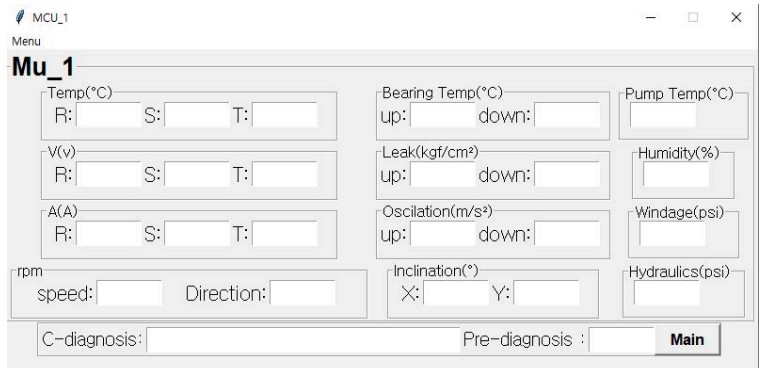

**Figure 4.** Monitoring system configuration screen of embedded system.

### 2.4. Monitoring of Server Computer

The sensor values received from the MCU are stored in the database and monitored on the screen in the server computer. A total of 12 types of sensors were configured, and the system was developed to be accessible from both computers and smartphones. Additionally, the submersible pump status can be displayed in three stages: normal, caution, and shutdown.

Figure 5 displays the monitoring screen connected to the computer and the operation status screen connected to the smartphone.

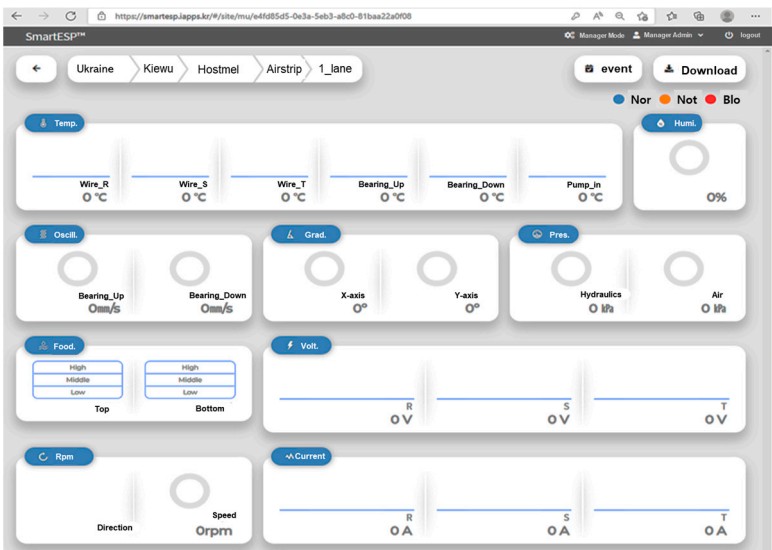

**Figure 5.** Monitoring system configuration screen of server computer.

## 3. Failure Diagnosis Algorithm

### 3.1. Individual Fault Diagnosis Algorithm

The fault diagnosis of the ICT convergence smart ESP was performed directly in the embedded system installed in the field, considering internet failures, among other factors [20–27]. The individual diagnosis of each sensor diagnosed the operating state of the ESP as normal, caution, or shut-off by consulting the technology of the Electrical Safety Research Institute, which is affiliated with the Korea Electrical Safety Corporation.

The fault diagnosis method consists of the following four steps.

1. Acquisition of sensor information from MCU.
2. Store the sensor information in the database.
3. Comparing individual sensor values with reference values to determine ESP status as normal, caution, or blocked.
4. The diagnosis result is transmitted to the MCU, and the MCU executes an action appropriate to the situation. (In case of blocking, a relay operation is executed.)

Table 2 shows the threshold values used for the individual diagnosis of ESP motion sensors. Here, the normal state refers to the case except for attention and blocking.

**Table 2.** Diagnostic criteria for single sensor.

| Measurement Factor | Division | Level | Sensor Value |
|---|---|---|---|
| Voltage | over | block | 380 V (+10% or more) |
| | | caution | 380 V (+5%~less than +10%) |
| | low | block | 380 V (−20% or less) |
| | | caution | 380 V (−10%~less then −20%) |
| Electric current | over | block | 30 A (+20% or more) |
| | | caution | 30 A (+10%~less than +20%) |
| Temperature (Winding) | high | block | 100 °C (+20% or more) |
| | | caution | 100 °C (+5~less than +20) |
| Temperature (Bearing) | high | block | 100 °C or more |
| | | caution | 80 °C more less than 100) |
| Humidity | high | block | 80% or more |
| | | caution | 70~less than 80% |
| Direction of rotation | reverse | block | in the negative direction |
| Oscillation | - | block | average (+20% or more) |
| | | caution | average (+10~less than +20%) |
| Gradient | - | block | {$x$-axis ($\pm10°$ or more)} or {$y$-axis ($\pm10°$ or more) |
| | | caution | {$x$-axis ($\pm5$~less then $\pm10°$)} or {$y$-axis ($\pm5$~$\pm10°$ or more)} |
| Flooding | - | block | High (Level-3, 4) or more |
| | | caution | Middle (Level-2) |
| Pressure (Oil/Air) | - | block | average ($\pm10\%$ or more) |
| | | caution | average (+5%~less than +10%) |

### 3.2. Complex Fault Diagnosis Algorithm

Complex fault diagnosis is performed by combining values from individual sensors, with a minimum of two and a maximum of five sensor values. The difference between a complex diagnosis and an individual diagnosis is that the former utilizes sensor information a total of 12 times by combining various sensors, rather than solely relying on the current value of a single sensor. For complex diagnosis, the diagnostic value was determined based on the advice from the Electrical Safety Research Institute, resulting in four warnings and eight blocks. Table 3 presents the diagnostic algorithms used for a complex diagnosis. In the composite diagnosis determination method, all sensor values shown in Table 3 are diagnosed under the AND condition.

**Table 3.** Judgment Criteria for Composite Diagnosis.

| Diagnosis Type | | Level | Voltage | Current | Sensor | | | |
| --- | --- | --- | --- | --- | --- | --- | --- | --- |
| | | | | | Temperature | Speed | Oscillation | Pressure (Oil) |
| Power | misconnection | block | 380 V ± 10% less | 30 A + 5% less | - | −720 rpm (±5% less) | - | - |
| | phase loss | block | 1~2 phase voltage 0 V | 1~2 line current 0 A | - | 720 rpm (±50% more) | - | - |
| Motor | winding open | block | 380 V ± 10% less | 1~2 line current 0 A | - | 720 rpm (±50% more) | - | |
| | inter layer short circuit | block | 380 V ± 10% less | Two currents are reduced by more than 10% than the average current. | - | - | - | |
| | | caution | 380 V ± 10% less | The two currents are reduced by 5~10% less than the average current. | - | - | - | |
| | rotor failure | block | 380 V ± 10% less | 1 current decreases by more than 10% than the average current. | - | - | average 15% more | |
| | | caution | 380 V ± 10% less | 1 current is reduced by 5% to 10% less than the average current. | - | - | average 10~15% more | |
| | bearing failure | block | 380 V ± 10% less | 30 A + 5% less | Bearing over 90 °C | - | average (15% more) | average |
| | | caution | 380 V ± 10% less | 30 A + 5% less | Bearing less than 80~90 °C | - | average (10~15% less) | average |
| | oil loss | block | 380 V ± 10% less | 30 A + 5% less | - | - | - | average 10% less |
| | | caution | 380 V ± 10% less | 30 A + 5% less | - | - | - | average 5~10% less |
| ESP | stuck | data | 380 V ± 10% less | 30 A × 1.1 more | - | 720 rpm × 0.9 less | average (15% more) | - |

### 3.3. Failure Prediction Diagnosis Algorithm

3.3.1. Failure Prediction Process

In order to predict submersible pump failures, the process is largely classified into three steps: data construction, learning and parameter adjustment, and verification and prediction. Data construction was carried out by obtaining data from 23 individual sensors mounted on the submersible pump, which were then stored in the server computer database through the TCP/IP communication of the MCU. In the learning step, the structure of the neural network is determined by learning the non-linear relationship between input and output variables using the back propagation algorithm. In the verification step, prediction is attempted with the structure determined through learning and Equation (1), and the accuracy of failure prediction is verified by analyzing the performance error of the model [28–32].

$$\text{MSE} = \frac{1}{2} \sum_k (y_k - t_k)^2 \qquad (1)$$

here, $y_k$ is the output of the neural network, $t_k$ is the value of the actual data, and $k$ is the number of dimensions of the data.

### 3.3.2. Multi-Layer Perceptron (MLP) Neural Network

An artificial neural network is a model that can solve problems by changing the synaptic coupling strength through the learning of artificial neurons (nodes) that form a network by synaptic coupling. There are several structures in the human neural network model, and one of them is the perceptron, which consists of an input layer, an output layer, weights, and threshold vectors. The feature of the perceptron is that the activation function of the hidden layer returns only values between $-1$ and $1$ as an output. Therefore, in order to apply the sensor value as an input variable to the activation function in the hidden layer, it is necessary to preprocess the data so that the value of each sensor falls between 0 and 1, as shown in Equation (2).

$$I_i = \frac{X - X_{min}}{X_{max} - X_{min}} \tag{2}$$

here, $I_i$ is a normalized value through standardization, $X$ : measured sensor value, $X_{min}$ : measured sensor minimum value, $X_{max}$ : measured sensor maximum value.

The normalized input value is sent to the activation function, and the input data of the hidden layer is calculated as shown in Equation (3). In this paper, the sigmoid activation function was used.

$$H_d = f_{(I_i)} \tag{3}$$

where $H_d$ is the hidden layer input data and $f$ is the activation function.

In the case of the hidden layer, it is calculated as in Equation (4) by multiplying the input data of the hidden layer calculated by the activation function by the weight and adding the threshold value. The calculated value is sent to the output layer.

$$y = f\left\{\sum(w_{kj} + H_d) + B\right\} \tag{4}$$

here, y is the output of the neural network, $w_{kj}$ is the weight of the hidden layer, $B$ is the threshold of the hidden layer, and f is the activation function.

As such, the multi-layer perceptron has a similar structure to the single-layer perceptron, but is composed of several layers, and the limitations of models consisting of one coordination layer in the single-layer perceptron and the linear separation problem are solved through the backpropagation algorithm.

In this paper, a multi-layer perceptron that overcomes the disadvantages of a single-layer perceptron is selected, and the applicability of failure diagnosis prediction of submersible pumps is evaluated by using a backpropagation algorithm method suitable for nonlinear prediction.

Figure 6 shows the composition of the training data of the MLP model. The values of the individual sensors were stored in a database and then randomly separated at a ratio of 6:1:3 for training, validation, and testing. The validation data was used to check that the model did not overfit the training data during the training process and was not used for model training. The test data are used to evaluate the performance of the trained model.

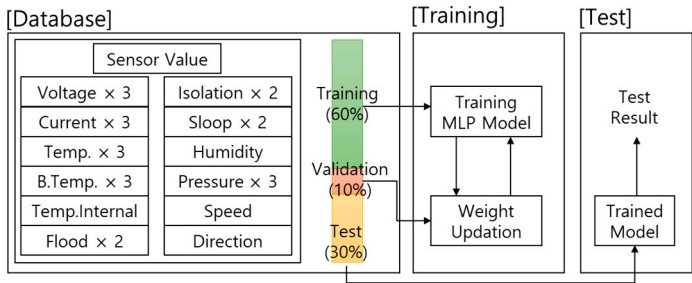

**Figure 6.** Data set configuration and MLP model training, validation, and test process.

### 3.3.3. Network Design

In this study, the MLP model was utilized to predict submersible pump failures.

The input layer of the MLP model consisted of 23 sensor values of the submersible pump, and the number of neurons in the input layer was also set to 23. The output layer was composed of one neuron for failure prediction. The number of neurons in the hidden layer was determined by testing different numbers of neurons, as shown in Table 4. Table 4 presents the accuracy measurements for the different numbers of hidden neurons, ranging from 2 to 64. The results showed that the highest accuracy was achieved with 16 neurons in the hidden layer.

**Table 4.** Tests for Determining Neurons in the Hidden Layer.

| Number of Neurons in the Hidden Layer | Test Set Accuracy | Number of Neurons in the Hidden Layer | Test Set Accuracy |
|---|---|---|---|
| 2 | 0.436333 | 20 | 0.949000 |
| 4 | 0.798333 | 30 | 0.938500 |
| 8 | 0.911167 | 32 | 0.927667 |
| 10 | 0.949000 | 40 | 0.914167 |
| 16 | 0.963833 | 64 | 0.850000 |

In this paper, the MLP network was determined based on the test results of the hidden layer. Figure 7 shows the proposed diagnosis algorithm, where the input layer utilized a total of 23 inputs from individual sensors, and the hidden layer was composed of 16 neuron layers, and the gradient descent method was applied for learning.

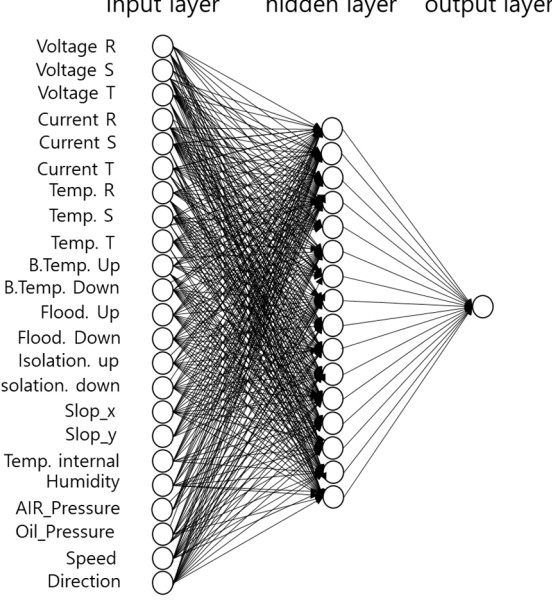

**Figure 7.** ESP's failure predictive diagnostic neural network.

In the failure prediction diagnosis, the individual sensor values obtained by the monitoring unit were used to determine the optimal parameters using the training data, and then the model was evaluated using the test data. A value of 0.8 or higher among the resulting values obtained by applying the MLP was set to be considered as indicating a high probability of failure.

## 4. Simulation and Discussion

### 4.1. Simulation of the Proposed Method

The proposed ICT convergence smart ESP was installed and tested indoors prior to operation. The MU was directly manufactured on the PCB and mounted on the existing

ESP, while the MCU was fixed on the outside of the ESP to handle central operations, communication, data storage, TCP/IP communication, monitoring, and individual and complex diagnosis in a 380 V–25 A environment. Figure 8 depicts the ICT convergence smart ESP, which includes both manufactured products and programs.

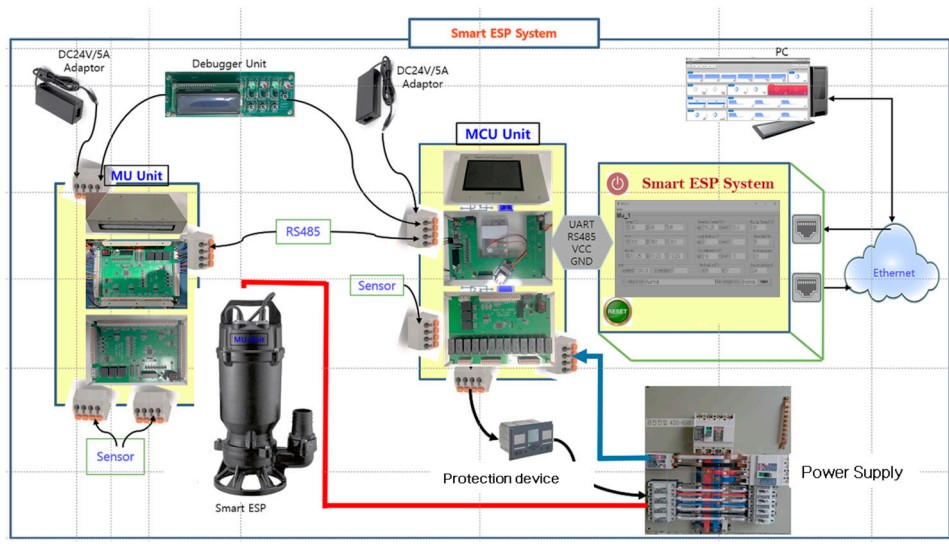

**Figure 8.** Behavioral test of ESP.

Figure 9 shows the voltage, current, temperature, vibration, and diagnosis results of the proposed ICT convergence smart ESP.

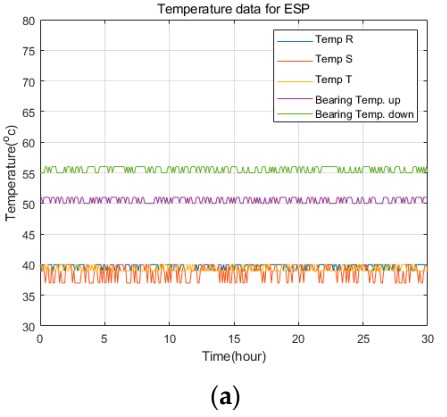

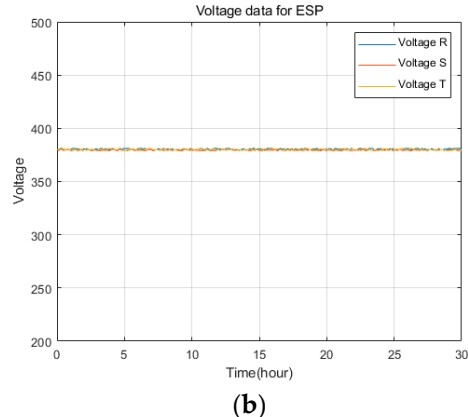

(**a**)    (**b**)

**Figure 9.** *Cont*.

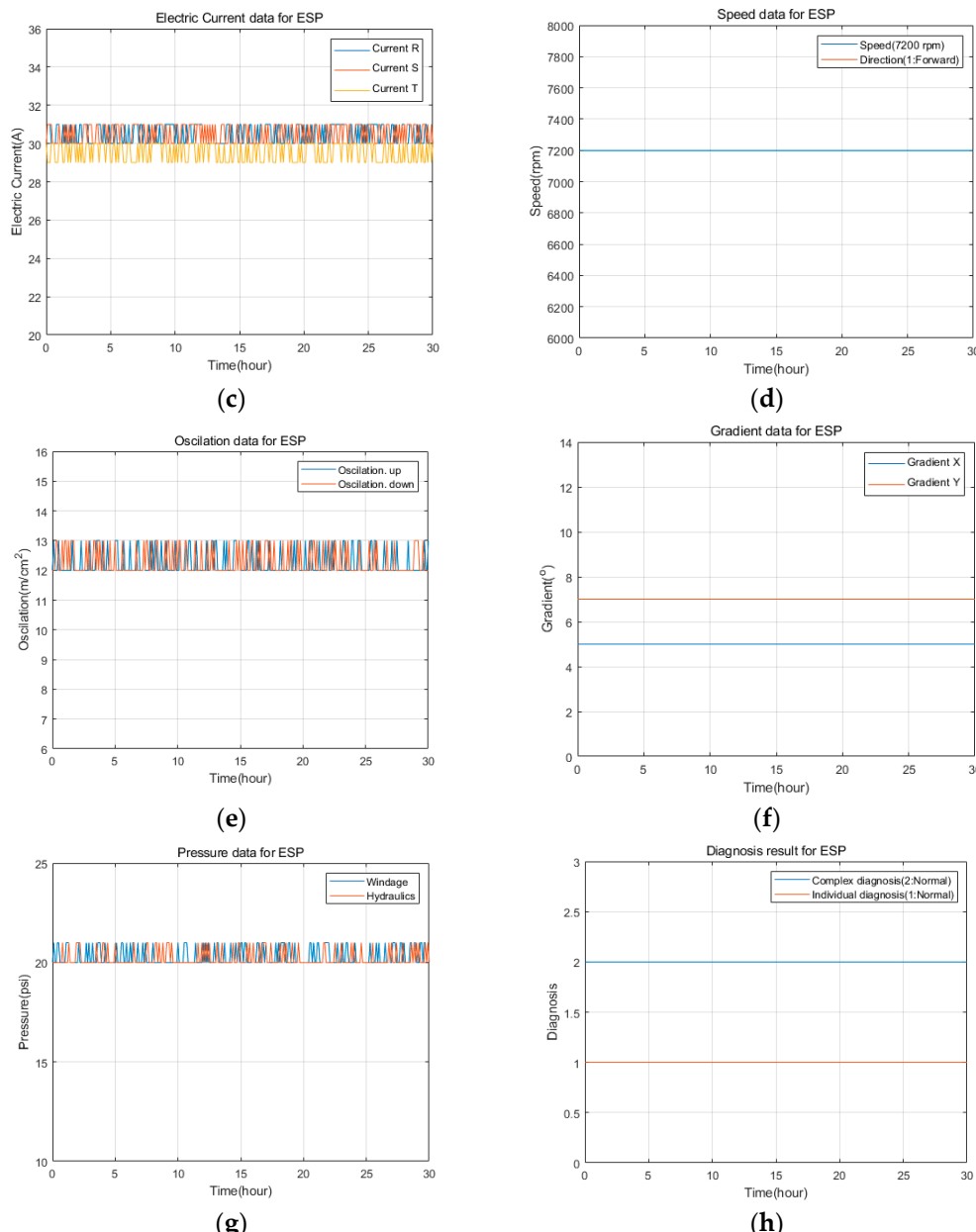

**Figure 9.** ESP operation state: (**a**) Temperature; (**b**) Voltage; (**c**) Current; (**d**) Speed/direction; (**e**) Oscillation; (**f**) Gradient; (**g**) Diagnosis result. (**h**) Diagnosis result.

Figure 9a shows the measured temperature. Data were collected by installing temperature sensors on the upper and lower parts of the three-phase power supply and bearing, respectively, to measure the temperature. As a result, 38~41 [°C] for all three phases, 55[c] at the upper part of the bearing, and 51[c] at the lower part were measured. Figure 9b shows the three-phase voltage, and the measurement result was confirmed to be 376~383 [V]. Figure 9c shows the three-phase current, and the measurement result was confirmed to be 28~32 [A]. Figure 9d shows the speed and direction of the motor. The motor speed was 720 rpm and it operated normally in the forward direction. Figure 9e is the result of measuring the vibration sensors installed on the top and bottom of the motor. The upper and lower motor vibration values were measured at 12 to 14 [m/cm$^2$]. Figure 9f is the measurement result of the tilt sensor installed on the motor. The x value of the slope was measured at 5° and they value at 7°. Figure 9g is the result of measuring the pressure sensor. Both hydraulic and air pressures were measured at 20 to 22 [psi].

Figure 9h shows the results of individual diagnosis and composite diagnosis. Individual diagnosis is judged normal when each sensor value is within the normal range. The composite diagnosis is determined based on the values in Table 3, and is determined to be normal.

The MCU transmits data received from the MU to the embedded system via RS485 communication and to the server computer via TCP/IP communication. A monitoring program is written in Python for the embedded system, which performs individual and complex diagnostics and transmits the results to the MCU and server computer. The server computer stores the received data in the database and writes a monitoring program using C#. Figure 10 shows the ESP operation status on the screen. Figure 10a shows the case of the normal operation, and Figure 10b shows the fault state where no current is supplied.

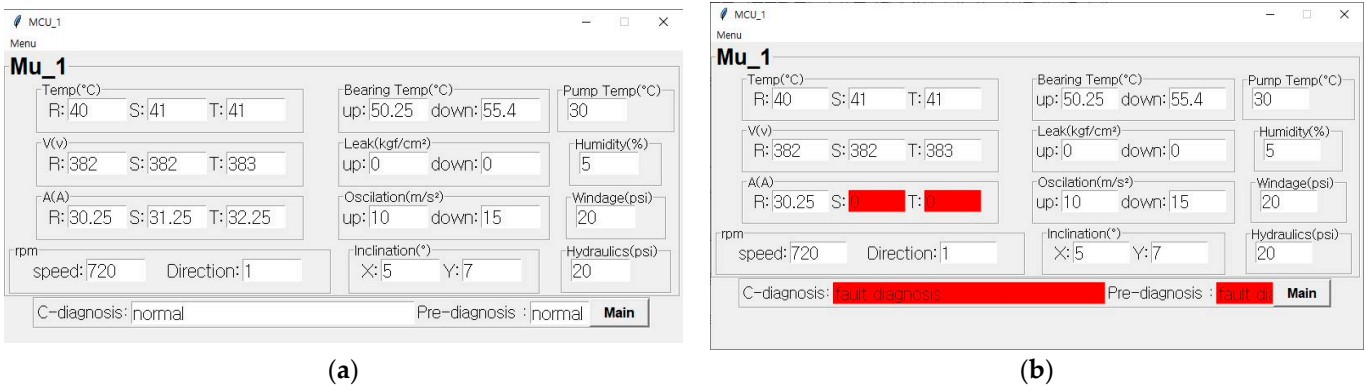

(**a**)　　　　　　　　　　　　　　　　　　　　　　　　(**b**)

**Figure 10.** Monitoring status of ESP: (**a**) Normal operation monitoring; (**b**) Fault operation monitoring.

Figure 11 shows the output values obtained by applying MLP to predict the failure of the submersible pump. All the sensor values of the current were within the normal range, and the diagnosis was normal for failure prediction. The number of iterations was 2000, the overall learning error was 0.033%, and the failure diagnosis prediction rate was 98.84%.

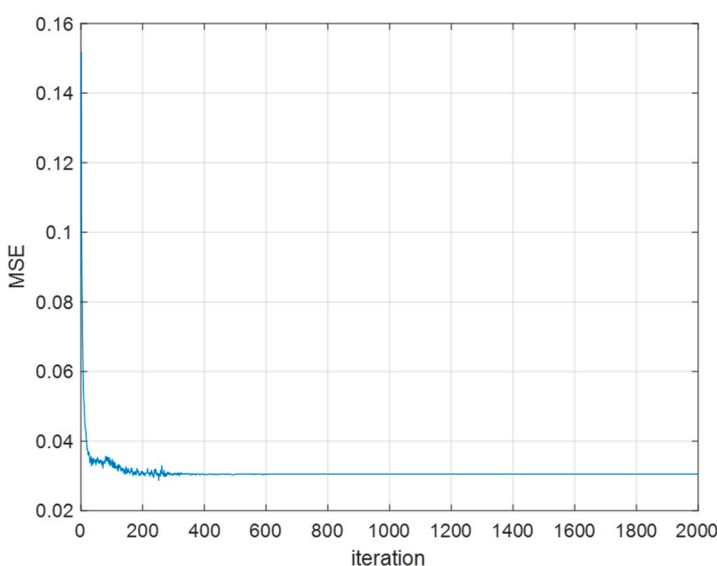

**Figure 11.** MSE value according to the number of repetitions applying MLP.

### 4.2. Comparison of Simulation Results

The simulation results compared the proposed method with existing methods such as SVM (support vector machine), LR (logistic regression), and RNN (recurrent neural network). The hidden layer of the RNN was simulated with the same layer size (16) as the proposed method.

Figure 12 presents the simulation results, which confirmed that RNN, SVM, the proposed method, and LR were excellent in order.

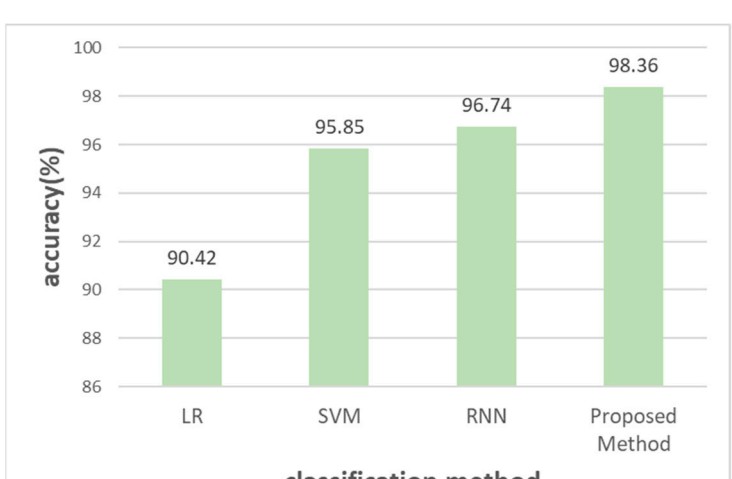

**Figure 12.** The performance of all methods.

## 5. Conclusions

This paper proposes a method for monitoring, diagnosing, and predicting failures in submersible pumps. The method combines a server computer, an embedded system, and a microprocessor to implement the monitoring and diagnosis. The submersible pump monitoring unit gathers values from 23 sensors mounted on the pump and sends the data to the monitoring control unit through serial communication. The MCU converts the sensor data into actual values, which are then transmitted to the server computer and embedded system. If a fault diagnosis system issues a stop command, the submersible pump stops its operation via a relay. In the embedded system, the sensor values are stored in a database and individual and complex diagnoses, as well as failure predictions, are performed. The maximum diagnostic time was achieved by applying MLP, using a total of 23 inputs from the individual sensors and a hidden layer that classifies complex diagnoses. The results are then sent to the MCU and server computer. The server computer is web-based, allowing for remote monitoring and the checking of sensor values via a smartphone. The results show that the proposed method confirms the operation status of the submersible pump, and enables both a failure diagnosis and prediction. It is believed that this method can increase competitiveness in the international market compared to existing products.

**Funding:** This paper was supported by Wonkwang University in 2022.

**Conflicts of Interest:** The authors declare no conflict of interest.

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
