# Peer review of "A Study on the Development of ICT Convergence Smart ESP Using Embedded System"

_electronics, doi:10.3390/electronics12061351_

Round 1

Reviewer 1 Report (Previous Reviewer 1)

Major:
Input parameters (23 of them) should be analysed in the following way:
1. remove particular parameter
2. train network
3. check quality of network
4. if quality is high - it means that parameter is not significant
repeat above steps for all parameters

remove marked parameters and train network again

Minors:

Table 1:
Missing horizontal ruler: Oscillation/Humidity

Fig.3
Larger image required

Table 3:
*1.1 <- \cdot 1.1
*0.9 <- \cdot 0.9
Data <-data

Eq.4
larger brackets required

Figure 6
Please change \ to \cross
Meaning of colors is required

Author Response

Review 1

The completion of our thesis seems to have improved due to the faithful review of the judges. The points pointed out by the judges were revised and supplemented as follows and reflected 100% in the original paper. Thank you.

Major:

Input parameters (23 of them) should be analysed in the following way:

  1. remove particular parameter
  2. train network
  3. check quality of network
  4. if quality is high - it means that parameter is not significant

repeat above steps for all parameters

remove marked parameters and train network again

 Answer : In this paper, the input parameters are characterized by monitoring the operation of the submersible pump, such as voltage and current. Individual sensors can confirm normal and anomalous operation of submersible pumps. Therefore, in this paper, we opted not to focus on the input parameters, but rather on determining the optimal number of hidden layers through learning.

Minors:

  1. Table 1: Missing horizontal ruler: Oscillation/Humidity

 Answer : It was reflected in the revised paper line [102]

  1. Fig.3 Larger image required

 Answer : It was reflected in the revised paper line [117]

  1. Table 3: *1.1 <- \cdot 1.1, *0.9 <- \cdot 0.9, Data <-data

 Answer : It was reflected in the revised paper line [179]

  1. Eq.4 larger brackets required

 Answer : It was reflected in the revised paper line [215]

Figure 6 Please change \ to \cross, Meaning of colors is required

 Answer : It was reflected in the revised paper line [234], Colors represent the percentage ratio.

Reviewer 2 Report (New Reviewer)

This paper develops a condition monitoring system based on embedded system and ICT (Information and Communication Technologies) technology for fault diagnosis and prediction of an electrical submersible pump. The proposed method is implemented by combination and communication of a server computer, an embedded system, and a microprocessor. The used technique is multi-layer perceptron for fault prediction. In summary, this paper is not sufficiently innovative in terms of algorithms. There is insufficient experimental validation and no comparative study proving the advantages of the method.

The major comments are given as follows,

1.      Technically speaking, the contribution and innovation should be clarified. Fault diagnosis and fault prediction are hot topics that have received extensive attention, “A Review of Real-Time Fault Diagnosis Methods for Industrial Smart Manufacturing, DOI: 10.3390/pr11020369”. What is the advantage of this work compared to other methods?

2.      The experimental validation is also insufficient. For example, the experiments in Section 4 are all normal states and lack the presentation of diagnosis results for fault states.

3.      The results on fault prediction only show the MSE metrics in the normal state, which is obviously not sufficient. More fault prediction results and comparative experiments should be analyzed.

4.      Only the temperature, voltage, current, speed, and oscillation data are shown in Fig.9, not all the measurement factors in Tables 2 and 3. What exactly are the factors that affect the diagnostic results should be given.

5.      How the values in Fig. 9 (f) and the corresponding diagnostic results were derived need to be further analyzed.

6.      Check the captions of Fig. 1 and Table 1.

7.      Check that the labels of references and figures are correctly cited.

8.      Some figures are blurry, such as Fig. 5 and Fig. 8.

9.      The explanation of Eq. (1) is not sufficient, what is E?

10.   This paper is written terribly, too many typos and writing errors, including language and format. The author should check the writing carefully.

Author Response

Review 2.

The completion of our thesis seems to have improved due to the faithful review of the judges. The points pointed out by the judges were revised and supplemented as follows and reflected 100% in the original paper. Thank you.

The major comments are given as follows,

  1. Technically speaking, the contribution and innovation should be clarified. Fault diagnosis and fault prediction are hot topics that have received extensive attention, “A Review of Real-Time Fault Diagnosis Methods for Industrial Smart Manufacturing, DOI: 10.3390/pr11020369”. What is the advantage of this work compared to other methods?

 Answer : First of all, thank you for recommending the related papers to this paper. After reviewing the above paper, it seems to be a paper that explains the overall research status of faults and diagnosis rather than direct research. The features of the submitted paper are the utilization of individual diagnosis and composite diagnosis, and it is possible to predict faults with these two features.

  1. The experimental validation is also insufficient. For example, the experiments in Section 4 are all normal states and lack the presentation of diagnosis results for fault states.

 Answer : Section 4 further reflects the results of failure conditions. It was reflected in the revised paper line [296]

  1. The results on fault prediction only show the MSE metrics in the normal state, which is obviously not sufficient. More fault prediction results and comparative experiments should be analyzed.

 Answer : There are methods for analyzing performance errors in neural networks, such as Mean Squared Error (MSE), Binary Cross Entropy (BCE), Confusion Matrix, Receiver Operating Characteristic (ROC) Curve, and Precision-Recall Curve. Among these, Mean Squared Error (MSE) is the most commonly used method for regression problems where the aim is to predict continuous values and it is also used in this paper.

  1. Only the temperature, voltage, current, speed, and oscillation data are shown in Fig.9, not all the measurement factors in Tables 2 and 3. What exactly are the factors that affect the diagnostic results should be given.

 Answer : All measurement elements from Tables 2 and 3 have been incorporated into Figure 9. It was reflected in the revised paper line [286]

  1. How the values in Fig. 9 (f) and the corresponding diagnostic results were derived need to be further analyzed.

 Answer : It was reflected in the revised paper line [281-284]

  1. Check the captions of Fig. 1 and Table 1.

 Answer : It was reflected in the revised paper line [76, 102]

  1. Check that the labels of references and figures are correctly cited.

 Answer : The revised manuscript reflected the revisions of the reviewers.

  1. Some figures are blurry, such as Fig. 5 and Fig. 8.

 Answer : It was reflected in the revised paper line [141, 265]

  1. The explanation of Eq. (1) is not sufficient, what is E?

 Answer : Equation (1) represents the squared error between the predicted values of the neural network model and the actual values. "E" represents the Mean Squared Error (MSE). It was reflected in the revised paper line [191]

  1. This paper is written terribly, too many typos and writing errors, including language and format. The author should check the writing carefully.

 Answer : The revised manuscript reflected the revisions of the reviewers.

Reviewer 3 Report (New Reviewer)

1.The introduction to the subject of the article was developed by the author at a good level.

2. In the article submitted for review, the author included only 31 scientific items in the bibliography - articles and monographs. All items presented in the bibliography are up to date. The oldest of them comes from 2005, (no. 7 Alfayez,L.; Mba,D.; Dyson,D.; The application of acoustic emission for detecting incipient cavitation and the best efficiency
point of a 60kW centrifugal pump: Case study, Ndt & E International,2005, 38,5, 354-358.)
which proves that the author is responsible for all research and results from the content of the presented article.

3. The article lacks an elaboration - a critical review concerning the content presented in the articles from the bibliography and what the author wanted to present in his research. I would like to ask the author to briefly present such content in the article.

4. We receive articles for review where co-authors are min. 4 people, sometimes more. Therefore, kudos to the author of the article for the independent development of the entire article.

5. In my opinion, the graphs are too small, there is a lot of content on them that the Author wanted to present, and therefore some of them are illegible - see fig. 5 and 10 b) - I would like to ask you to improve these drawings because they are difficult to read.

6. Please extend conclusions that are too short for such an article. They contain only 9 general sentences, without reference to what the author himself developed during his research.

7. Is it possible to change the scale on the variable waveforms Fig. 9 a) to f), especially on the OY axis, not to present data from 0 but e.g. 30 - 60 as in Fig. 9 a). 30 was the beginning on the OY axis - the waveforms would be visible better, or another solution - use a magnifying glass on the graph - magnification in a short time 10 - 15 h. To the author's decision.

8. An interesting article, if the author corrects minor remarks, I recommend it for printing.

9. The development of a critical review of the literature will enable the author of the article to significantly expand the bibliography.

Author Response

Review 3

The completion of our thesis seems to have improved due to the faithful review of the judges. The points pointed out by the judges were revised and supplemented as follows and reflected 100% in the original paper. Thank you.

1.The introduction to the subject of the article was developed by the author at a good level.

 Answer : Thank you for reviewing the review. The revised manuscript reflected the revisions of the reviewers.

  1. In the article submitted for review, the author included only 31 scientific items in the bibliography - articles and monographs. All items presented in the bibliography are up to date. The oldest of them comes from 2005, (no. 7 Alfayez,L.; Mba,D.; Dyson,D.; The application of acoustic emission for detecting incipient cavitation and the best efficiency

point of a 60kW centrifugal pump: Case study, Ndt & E International,2005, 38,5, 354-358.) which proves that the author is responsible for all research and results from the content of the presented article.

 Answer : Thank you for reviewing the review. The revised manuscript reflected the revisions of the reviewers.

  1. The article lacks an elaboration - a critical review concerning the content presented in the articles from the bibliography and what the author wanted to present in his research. I would like to ask the author to briefly present such content in the article.

 Answer : Some parts of the introduction and conclusion have been revised. The revised manuscript reflected the revisions of the reviewers.

  1. We receive articles for review where co-authors are min. 4 people, sometimes more. Therefore, kudos to the author of the article for the independent development of the entire article.

 Answer : Thank you for reviewing the review. The revised manuscript reflected the revisions of the reviewers.

  1. In my opinion, the graphs are too small, there is a lot of content on them that the Author wanted to present, and therefore some of them are illegible - see fig. 5 and 10 b) - I would like to ask you to improve these drawings because they are difficult to read.

 Answer : It was reflected in the revised paper line [141, 296]

  1. Please extend conclusions that are too short for such an article. They contain only 9 general sentences, without reference to what the author himself developed during his research.

 Answer : Some content has been added and the conclusion has been re-written. It was reflected in the revised paper line [307-323]

  1. Is it possible to change the scale on the variable waveforms Fig. 9 a) to f), especially on the OY axis, not to present data from 0 but e.g. 30 - 60 as in Fig. 9 a). 30 was the beginning on the OY axis - the waveforms would be visible better, or another solution - use a magnifying glass on the graph - magnification in a short time 10 - 15 h. To the author's decision.

 Answer : It was reflected in the revised paper line [286]

  1. An interesting article, if the author corrects minor remarks, I recommend it for printing.

 Answer : Thank you for reviewing the review. The revised manuscript reflected the revisions of the reviewers.

  1. The development of a critical review of the literature will enable the author of the article to significantly expand the bibliography.

 Answer : Thank you for reviewing the review. The revised manuscript reflected the revisions of the reviewers.

Round 2

Reviewer 2 Report (New Reviewer)

Some modifications have been made by authors, and the manuscript has been improved. While there still are several comments not considered. 

(1) I suggest that the authors carefully revise it and supply more  comparative experiments with other detection methods.

(2) Moreover the literatures analysis should be given in  introduction.  "In conclusion, recent research has focused on acquiring sensor values for voltage,  current, and operating characteristics, and then filtering the values to perform fault diag-59 nosis [10-32]." This kind of statement is inappropriate.

Author Response

Review 2.

The completion of our thesis seems to have improved due to the faithful review of the judges. The points pointed out by the judges were revised and supplemented as follows and reflected 100% in the original paper. Thank you.

(1) I suggest that the authors carefully revise it and supply more comparative experiments with other detection methods.

 Answer : It was reflected in the revised paper line [309~320]

(2) Moreover the literatures analysis should be given in introduction. "In conclusion, recent research has focused on acquiring sensor values for voltage, current, and operating characteristics, and then filtering the values to perform fault diag-59 nosis [10-32]." This kind of statement is inappropriate.

 Answer : It was reflected in the revised paper line [58~65]

This manuscript is a resubmission of an earlier submission. The following is a list of the peer review reports and author responses from that submission.

Round 1

Reviewer 1 Report

Major:

MLP is used for diagnosis but the approach from technical and scientical point-of-view is poor.

1. "23 input layers were used" <- there is single layer and 23 inputs of the input layer
2. Hidden layer uses 12 neurons, but what about testing of different configuration ?
Too much neurons - problems with interpolation. Too enough neurons - problems with solving problem.
It must be considered in tests.

3. There is no information about activation function used in neurons.
4. Lack of scaling of data before input
5. No tests related to required inputs. All data is input without any consideration as to whether it is actually needed.
6. Testing of neural network is not considered.

This is very, very week neural network.

Minors:

Fig.2
Please use \cross instead of '*' for the specification of number of lines

R,S,T markings of windings is used in some countries, please add (L1,L2,L3) as an alternative markings in l.103

Table.1
volt <- Volt

Table.1, Table.3
Please add horizontal lines between rows - too much mixed content in tables now

l.108 Please fix it:
"This is a table. Tables should be placed in the main text near to the first time they are cited."

Types or family of processors it also valuable information for reader - please add it.

l.152
etc <- etc.

l.215
[v] <- [V]  and more similar cases in this paper

l.225
"486 communication" ???
RS485 ???

Author Response

The completion of our thesis seems to have improved due to the faithful review of the judges. The points pointed out by the judges were revised and supplemented as follows and reflected 100% in the original paper. Thank you.

Major:

MLP is used for diagnosis but the approach from technical and scientical point-of-view is poor.

  1. "23 input layers were used" <- there is single layer and 23 inputs of the input layer

Answer : The 23 sensor values of the submersible pump are scaled and used as input.

  1. Hidden layer uses 12 neurons, but what about testing of different configuration ?

Too much neurons - problems with interpolation. Too enough neurons - problems with solving problem. It must be considered in tests.

Answer : Hidden layer use 12 neurons are the values used for complex diagnosis.

  1. There is no information about activation function used in neurons.

Answer :[212 line] The activation function used in this paper is the signoid function.

  1. Lack of scaling of data before input

Answer :[208 line] The scaling of the input data is as shown in Equation (2)

  1. No tests related to required inputs. All data is input without any consideration as to whether it is actually needed.

Answer : [234 line] The pointed out contents were added and reflected in the thesis.

  1. Testing of neural network is not considered.

Answer : [234 line] The pointed out contents were added and reflected in the thesis.

This is very, very week neural network.

Minors:

Fig.2

  1. Please use \cross instead of '*' for the specification of number of lines

Answer : [98 line] The pointed out contents were added and reflected in the thesis.

  1. R,S,T markings of windings is used in some countries, please add (L1,L2,L3) as an alternative markings in l.103

Answer : [101,105 line] The pointed out contents were added and reflected in the thesis.

  1. Table.1 volt <- Volt

Answer : [107 line] The pointed out contents were added and reflected in the thesis.

  1. Table.1, Table.3

Please add horizontal lines between rows - too much mixed content in tables now

Answer : [107,181 line] The pointed out contents were added and reflected in the thesis.

  1. l.108 Please fix it:

"This is a table. Tables should be placed in the main text near to the first time they are cited.“

Answer : [106 line] The pointed out contents were added and reflected in the thesis.

  1. Types or family of processors it also valuable information for reader - please add it.

Answer : [108 line] The pointed out contents were added and reflected in the thesis.

  1. l.152 etc <- etc.

Answer : [151 line] The pointed out contents were added and reflected in the thesis.

  1. l.215 [v] <- [V] and more similar cases in this paper

Answer : [262 line] The pointed out contents were added and reflected in the thesis.

  1. l.225 "486 communication" ??? RS485 ???

Answer : [273 line] The pointed out contents were added and reflected in the thesis.

Reviewer 2 Report

In this work, the author proposes a diagnosis approach of Electrical Submersible Pumps which can be useful for condition and maintenance tasks. Please, read below the following points

-The contribution of the paper is not clear. The state of the art is not well defined. What has been done in the field? What is lacking in the research on ESP diagnosis that this paper is contributing to?

-The approach proposed must be described precisely in terms of the methods and procedures used. The author states that “In particular, it is suggested that the failure diagnosis of heavy pumps can be diagnosed through complex diagnosis rather than based on individual sensor values.”. Complex diagnosis means nothing to the reader in terms of what method is being used or proposed.

-If the hardware design is a contribution of this paper, then it should be explained in a more systematic way. What is different about this hardware from current standards in the field?

-The composite diagnosis is rule-based. How were these rules obtained? Did the author design these rules? Is this a contribution of the paper?

-The design of the neural network used for diagnosis is not explained. How were the hyperparameters selected? The training and testing experiments should be described in detail.

-The diagnosis approach proposed is far too simple to be considered a contribution to the field.

Minor comments:

· Acronyms should be described before being used. For example, ICT is used in the abstract before explaining what it means. 

· English proofreading is a must for this manuscript. There are serious flaws in the writing. Consider using shorter sentences with a well-defined structure. The entire manuscript should be revised in this sense.

Author Response

The completion of our thesis seems to have improved due to the faithful review ofthe judges. The points pointed out by the judges were revised and supplemented as follows and reflected 100% in the original paper. Thank you

In this work, the author proposes a diagnosis approach of Electrical Submersible Pumps which can be useful for condition and maintenance tasks. Please, read below the following points

-The contribution of the paper is not clear. The state of the art is not well defined. What has been done in the field? What is lacking in the research on ESP diagnosis that this paper is contributing to?

Answer : In this paper, we acquired the status information of the submersible pump, checked the status of the submersible pump based on the status information, and predicted failures. Both hardware and software have been performed, and the most difficult part of ESP diagnosis is failure prediction.

-The approach proposed must be described precisely in terms of the methods and procedures used. The author states that “In particular, it is suggested that the failure diagnosis of heavy pumps can be diagnosed through complex diagnosis rather than based on individual sensor values.”. Complex diagnosis means nothing to the reader in terms of what method is being used or proposed.

Answer : [169ine] The status information of existing submersible pumps was diagnosed using only individual sensor values. The proposed method is in performing complex diagnosis and failure prediction. Related information was presented in 3.2.

-If the hardware design is a contribution of this paper, then it should be explained in a more systematic way. What is different about this hardware from current standards in the field?

Answer : In this paper, both hardware and software were performed. Currently, there is no standard for sensor acquisition method for operation status information of submersible pumps. The difference from the existing method is that the operation status of the submersible pump can be checked both on-site and by smartphone. It has also been suggested to be able to perform predictive diagnostics.

-The composite diagnosis is rule-based. How were these rules obtained? Did the author design these rules? Is this a contribution of the paper?

Answer : [152 line] The complex diagnosis method was performed after receiving confirmation from the Korea Electrical Safety Corporation.

-The design of the neural network used for diagnosis is not explained. How were the hyperparameters selected? The training and testing experiments should be described in detail.

Answer : [182 line] It has been added to 3.3 of the text.

-The diagnosis approach proposed is far too simple to be considered a contribution to the field.

Answer : The proposed method is for checking the condition of submersible pumps and predicting failures. This method applies a complex diagnosis method rather than the existing single sensor method.

Minor comments:

  • Acronyms should be described before being used. For example, ICT is used in the abstract before explaining what it means.

Answer : [9 line] The pointed out contents were added and reflected in the thesis.

  • English proofreading is a must for this manuscript. There are serious flaws in the writing. Consider using shorter sentences with a well-defined structure. The entire manuscript should be revised in this sense.

Answer : I read the paper in its entirety and corrected the English proofreading.

Round 2

Reviewer 1 Report

Major:
Reviewer (from previous review):
Hidden layer uses 12 neurons, but what about testing of different configuration ?
Too much neurons - problems with interpolation. Too enough neurons - problems with solving problem. It must be considered in tests.

Answer : Hidden layer use 12 neurons are the values used for complex diagnosis.

Reviewer:
1.This is not the correct answer - testing of importance for inputs is mandatory
2.No answer about number of hidden neurons - it is mandatory. What about configuration with 2,4,8,10,16,20,32 neurons for example ?
Please add new section for analysis of both problems.

Minors:

1.
Eq.4 multiple mistakes in formula - please check it:
first plus sign
there is single output so k-index for y is not significant

2.
What about activation function for output ? It is not supported by formula 4 ?

Author Response

The completion of our thesis seems to have improved due to the faithful review of the judges. The points pointed out by the judges were revised and supplemented as follows and reflected 100% in the original paper. Thank you.

Major:

1.This is not the correct answer - testing of importance for inputs is mandatory

2.No answer about number of hidden neurons - it is mandatory. What about configuration with 2,4,8,10,16,20,32 neurons for example ?

Please add new section for analysis of both problems.

Answer : [page : 9]The pointed out points were reflected in this paper

Minors:

1.Eq.4 multiple mistakes in formula - please check it:

first plus sign

there is single output so k-index for y is not significant

Answer : [line : 217] The pointed out points were reflected in this paper

  1. What about activation function for output ? It is not supported by formula 4 ?

Answer : [line : 218] The pointed out points were reflected in this paper

Round 3

Reviewer 1 Report

ok